# ADVERSARIAL ATTACKS ON COPYRIGHT DETECTION SYSTEMS

## ABSTRACT

It is well-known that many machine learning models are susceptible to adversarial attacks, in which an attacker evades a classifier by making small perturbations to inputs. This paper discusses how industrial copyright detection tools, which serve a central role on the web, are susceptible to adversarial attacks. We discuss a range of copyright detection systems, and why they are particularly vulnerable to attacks. These vulnerabilities are especially apparent for neural network based systems. As proof of concept, we describe a well-known music identification method and implement this system in the form of a neural net. We then attack this system using simple gradient methods. Adversarial music created this way successfully fools industrial systems, including the AudioTag copyright detector and YouTube's Content ID system. Our goal is to raise awareness of the threats posed by adversarial examples in this space and to highlight the importance of hardening copyright detection systems to attacks.

## 1   INTRODUCTION

Machine learning systems are easily manipulated by adversarial attacks, in which small perturbations to input data cause large changes to the output of a model. Such attacks have been demonstrated on a number of potentially sensitive systems, largely in an idealized academic context, and occasionally in the real-world (Tencent, 2019; Kurakin et al., 2016; Athalye et al., 2017; Eykholt et al., 2017; Yakura & Sakuma, 2018; Qin et al., 2019).

*Copyright detection systems* are among the most widely used machine learning systems in industry, and the security of these systems is of foundational importance to some of the largest companies in the world. Despite their importance, copyright systems have gone largely unstudied by the ML security community. Common approaches to copyright detection extract features, called fingerprints, from sampled video or audio, and then match these features with a library of known fingerprints. Examples include YouTube's *Content ID*, which flags copyrighted material on YouTube and enables copyright owners to monetize and control their content. At the time of writing this paper, more than 100 million dollars have been spent on Content ID, which has resulted in more than 3 billion dollars in revenue for copyright holders (Manara, 2018). Closely related tools such as Google Jigsaw detect and remove videos that promote terrorism or jeopardized national security. There is also a regulatory push for the use of copyright detection systems; the recent EU Copyright Directive requires any service that allows users to post text, sound, or video to implement a copyright filter.

A wide range of copyright detection systems exist, most of which are proprietary. It is not possible to demonstrate attacks against all systems, and this is not our goal. Rather, the purpose of this paper is to discuss why copyright detectors are especially vulnerable to adversarial attacks and establish how existing attacks in the literature can potentially exploit audio and video copyright systems.

As a proof of concept, we demonstrate an attack against real-world copyright detection systems for music. To do this, we reinterpret a simple version of the well-known "Shazam" algorithm for music fingerprinting as a neural network and build a differentiable implementation of it in TensorFlow (Abadi et al., 2016). By using a gradient-based attack and an objective that is designed to achieve good transferability to black-box models, we create adversarial music that is easily recognizable to a human, while evading detection by a machine. With sufficient perturbations, our adversarial music

successfully fools industrial systems,[1] including the AudioTag music recognition service (AudioTag, 2009), and YouTube's Content ID system(Google, 2019).

## 2 WHAT MAKES COPYRIGHT DETECTION SYSTEMS VULNERABLE TO ATTACKS?

Work on adversarial examples has been focused largely on imaging problems, including image classification, object detection, and semantic segmentation (Szegedy et al., 2013; Goodfellow et al., 2014; Moosavi-Dezfooli et al., 2016; 2017; Shafahi et al., 2018; Xie et al., 2017; Fischer et al., 2017). More recently, adversarial examples have been studied for non-vision applications such as speech recognition (i.e., speech-to-text) (Carlini & Wagner, 2018; Alzantot et al., 2018; Taori et al., 2018; Yakura & Sakuma, 2018). Attacks on copyright detection systems are different from these applications in a number of important ways that result in increased potential for vulnerability.

First, digital media can be directly uploaded to a server without passing through a microphone or camera. This is drastically different from physical-world attacks, where adversarial perturbations must survive a data measurement process. For example, a perturbation to a stop sign must be effective when viewed through different cameras, resolutions, lighting conditions, viewing angles, motion blurs, and with different post-processing and compression algorithms. While attacks exist that are robust to these nuisance variables (Athalye et al., 2017), this difficulty makes even white-box attacks difficult, leaving some to believe that physical world attacks are not a realistic threat model (Lu et al., 2017a;b). In contrast, a manipulated audio file can be uploaded directly to the web without passing it through a microphone that may render perturbations ineffective.

Second, copyright detection is an *open-set* problem, in which systems process media that does not fall into any known class (i.e., doesn't correspond to any protected audio/video). This is different from the closed-set detection problem in which everything is assumed to correspond to a class. For example, a mobile phone application for music identification may solve a closed-set problem; the developers can assume that every uploaded audio clip corresponds to a known song, and when results are uncertain there is no harm in guessing. By contrast, when the same algorithm is used for copyright detection on a server, developers must solve the open-set problem; nearly all uploaded content is not copyright protected, and should be labeled as such. In this case, there is harm in "guessing" an ID when results are uncertain, as this may bar users from uploading non-protected material. Copyright detection algorithms must be tuned conservatively to operate in an environment where most content does not get flagged.

Finally, copyright detection systems must handle a deluge of content with different labels despite strong feature similarities. Adversarial attacks are known to succeed easily in an environment where two legitimately different audio/video clips may share strong similarities at the feature level. This has been recognized for the ImageNet classification task (Russakovsky et al., 2015), where feature overlap between classes (e.g., numerous classes exist for different types of cats/dogs/birds) makes systems highly vulnerable to untargeted attacks in which the attacker perturbs an object from its home class into a different class of high similarity. As a result, state of the art defenses for untargeted attacks on ImageNet achieve far lower robustness than classifiers for simpler tasks (Shafahi et al., 2019; Cohen et al., 2019). Copyright detection systems may suffer from a similar problem; they must discern between protected and non-protected content even when there is a strong feature overlap between the two.

## 3 TYPES OF COPYRIGHT DETECTION SYSTEMS

Fingerprinting algorithms typically work by extracting an ensemble of feature vectors (also called a "hash" in the case of audio tagging) from source content, and then matching these vectors to a library of known vectors associated with copyrighted material. If there are enough matches between a source sample and a library sample, then the two samples are considered identical. Most audio, image, and video fingerprinting algorithms either train a neural network to extract fingerprint features, or extract hand-crafted features. In the former case, standard adversarial methods lead to immediate

---

[1]Affected parties were notified before publication of this article.

susceptibility. In the latter case, feature extractors can often be re-interpreted and implemented as shallow neural networks, and then attacked (we will see an example of this below).

For video fingerprinting, one successful approach by Saviaga & Toxtli (2018) is to use object detectors to identify objects entering/leaving video frames. An extracted hash then consists of features describing the entering/leaving objects, in addition to the temporal relationships between them. While effective at labeling clean video, recent work has shown that object detectors and segmentation engines are easily manipulated to adversarially place/remove objects from frames (Wang et al., 2019; Xie et al., 2017).

Works such as Li et al. (2019) build "robust" fingerprints by training networks on commonly used distortions (such as adding a border, adding noise, or flipping the video), but do not consider adversarial perturbations. While such networks are robust against pre-defined distortions, they will not be robust against white-box (or even black-box) adversarial attacks.

Similarly, recent plagiarism detection systems such as Yasaswi et al. (2017) rely on neural networks to generate a fingerprint for a document. While using the deep feature representations of a document as a fingerprint might result in a higher accuracy for the plagiarism model, it potentially leaves the system open to adversarial attacks.

Audio fingerprinting might appear to be more secure than the domains described above because practitioners typically rely on hand-crafted features rather than deep neural nets. However, we will see below that even hand-crafted feature extractors are susceptible to attacks.

## 4 CASE STUDY: EVADING AUDIO FINGERPRINTING

We now describe a commonly used audio fingerprinting/detection algorithm and show how one can build a differentiable neural network resembling this algorithm. This model can then be used to mount black-box attacks on real-world systems.

### 4.1 AUDIO FINGERPRINTING MODELS

An acoustic fingerprint is a feature vector that is useful for quickly locating a sample or finding similar samples in an audio database. Audio fingerprinting plays a central role in detection algorithms such as Content ID. Therefore, in this section, we describe a generic audio fingerprinting model that will ultimately help us generate adversarial examples.

#### 4.1.1 IMPORTANT FINGERPRINTING GUIDELINES FROM SHAZAM

Due to the financially sensitive nature of copyright detection, there are very few publicly available fingerprinting models. One of the few widely used publicly known models is from the Shazam team (Wang et al., 2003). Shazam is a popular mobile phone app for identifying music. According to the Shazam paper, a good audio fingerprint should have the following properties:

- *Temporally localized*: every fingerprint hash is calculated using audio samples that span a short time interval. This enables hashes to be matched to a short sub-sample of a song.
- *Translation invariant*: fingerprint hashes are (nearly) the same regardless of where in the song a sample starts and ends.
- *Robust*: hashes generated from the original clean database track should be reproducible from a degraded copy of the audio.

#### 4.1.2 THE HANDCRAFTED FINGERPRINTING MODEL

The *spectrogram* of a signal, also called the short-time Fourier transform, is a plot that shows the frequency content (Fourier transform) of the waveform over time. After experimenting with various features for fingerprinting, Wang et al. (2003) chose to form hashes from the locations of spectrogram peaks. Spectrogram peaks have nice properties such as robustness in the presence of noise and approximate linear superposability.

In the next subsection, we build a shallow neural network that captures the key ideas of Wang et al. (2003), while adding extra layers that help produce transferable adversarial examples. In particular,

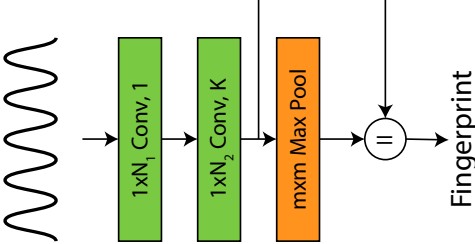

Figure 1: An audio fingerprinting model with two convolution layers and a max pooling layer. This model produces binary fingerprints by finding local maxima of the spectrogram of the input signal.

we add an extra smoothing layer that makes our model difficult to attack and helps us craft strong attacks that can transfer to other black-box models.

### 4.2 INTERPRETING THE FINGERPRINT EXTRACTOR AS A CNN

Here we describe the details of the generic neural network model we use for generating the audio fingerprints. Each layer of the network can be seen as a transformation that is applied to its input. We treat the output representation of our network as the fingerprint of the input audio signal. Ideally, we would like to extract features that can uniquely identify a signal while being independent of the exact start or end time of the sample. Convolutional neural networks maintain the *temporally localized* and *translation invariant* properties mentioned in section 4.1.1, and so we model the fingerprinting procedure using fully convolutional neural networks.

The first network layer convolves with a normalized Hann function, which is a filter of the form

$$f_1(n) = \frac{sin^2\left(\frac{\pi n}{N}\right)}{\sum_{i=0}^{N} sin^2\left(\frac{\pi i}{N}\right)}, \tag{1}$$

where $N$ is the width of the kernel. Convolving with a normalized Hann window smooths the adversarially perturbed audio waveform and the output of this layer is a perturbed but smooth audio sample that is then fingerprinted. This layer removes discontinuities and bad spectral properties that may be introduced into the signal during adversarial optimization and also makes the black-box attacks more efficient by preventing perturbations that do not transfer well to other models.

The next convolutional layer computes the spectrogram (aka Short Term Fourier Transform) of the waveform and converts the audio signal from its original domain to a representation in the frequency domain. This is accomplished by convolving with an ensemble of $N$ Fourier kernels of different frequencies, each with $N$ output channels. This convolution has filters of the form

$$f_2(k, n) = e^{-i2\pi kn/N}, \tag{2}$$

where $k \in 0, 1, \cdots, N-1$ is an output channel index and $n \in 0, 1, \cdots, N-1$ is the index of the filter coefficient. After this convolution is computed, we apply $|x|$ on the output to get the magnitude of the STFT.

After the convolutional layers, we get a feature representation of the audio signal. We call this feature representation $\phi(x)$, where $x$ is the input signal. This representation is susceptible to noise and a slight perturbation in the audio signal can change it. Furthermore, this representation is very dense which makes it relatively hard to store and search against all audio signals in the database. To address these issues, Wang et al. (2003) suggest using the local maxima of the spectrogram as features.

We can find local maxima within our neural net framework by applying a max pooling function over the feature representation $\phi(x)$. We then find the places where the output of the maxpool equals the original feature representation (i.e., the locations where $\phi(x) = maxpool\left(\phi(x)\right)$). The resulting binary map of local maxima locations is the fingerprint of the signal and can be used to search for a signal against a database of previously processed signals. We will refer to this binary fingerprint as $\psi(x)$ where $x$ is the input signal. Figure 1 depicts the 2-layer convolutional network we use in this work for generating signal fingerprints.

### 4.3 Formulating the adversarial loss function

To craft an adversarial perturbation, we need a differentiable surrogate loss that measures how well an extracted fingerprint matches a reference. The CNN described in section 4.2 uses spectrogram peaks to generate fingerprints, but we did not yet specify a loss for quantifying how close two fingerprints are. Once we have such a loss, we can use standard gradient methods to find a perturbation $\delta$ that can be added to an audio signal to prevent copyright detection. To ensure the similarity between perturbed and clean audio, we bound the perturbation $\delta$. That is, we enforce $\|\delta\|_p \leq \epsilon$. Here $\|.\|_p$ is the $\ell_p$-norm of the perturbation and $\epsilon$ is the perturbation budget available to the adversary. In our experiments, we use the $\ell_\infty$-norm as our measure of perturbation size.

The simplest similarity measure between two binary fingerprints is simply the Hamming distance. Since the fingerprinting model outputs a binary fingerprint $\psi(x)$, we can simply measure the number of local maxima that the signals $x$ and $y$ share by $|\psi(x) \cdot \psi(y)|$. To make a differentiable loss function from this similarity measure, we use

$$J(x,y) = |\phi(x) \cdot \psi(x) \cdot \psi(y)|. \tag{3}$$

In the white box case where the fingerprinting system is known, attacks using the loss (3) are extremely effective. However, attacks using this loss are extremely brittle and do not transfer well; one can minimize this loss by changing the locations of local maxima in the spectrogram by just one pixel. Such small changes in the spectrogram are unlikely to transfer reliably to black-box industrial systems.

To improve the transferability of our adversarial examples, we propose a robust loss that promotes large movements in the local maxima of the spectrogram. We do this by moving the locations of local maxima in $\phi(x)$ outside of any neighborhood of the local maxima of $\phi(y)$. To efficiently implement this constraint within a neural net framework, we use two separate max pooling layers, one with a bigger width $w_1$ (the same width used in fingerprint generation), and the other with a smaller width $w_2$. If a location in the spectrogram yields output of the $w_1$ pooling strictly larger than the output of the $w_2$ pooling[2], we can be sure that there is no spectrogram peak within radius $w_2$ of that location.

Equation 4 describes a loss function that penalizes the local maxima of $x$ that are in the $w_2$ neighborhood of local maxima of $y$. This loss function forces the output of the max pooling layers to be different by at least a margin $c$.

$$J(x,y) = \sum_i \left( ReLU \left( c - \left( \max_{|j| \leq w_1} \phi(i+j;x) - \max_{|j| \leq w_2} \phi(i+j;x) \right) \right) \cdot \psi(i;y) \right) \tag{4}$$

Finally, we make our loss function differentiable by replacing the maximum operator with the smoothed max function

$$S_\alpha(x_1, x_2, \cdots, x_n) = \frac{\sum_{i=1}^n x_i e^{\alpha x_i}}{\sum_{i=1}^n e^{\alpha x_i}}, \tag{5}$$

where $\alpha$ is a smoothing hyper parameter. As $\alpha \to \infty$, the smoothed max function more accurately approximates the exact max function. For simplicity, we chose $\alpha = 1$ for all experiments.

### 4.4 Crafting the evasion attack

We solve the bounded optimization problem

$$\min_\delta J(x+\delta, x) \qquad s.t. \|\delta\|_\infty \leq \epsilon, \tag{6}$$

where $x$ is the benign audio sample, and $J$ is the loss function defined in equation 4 with the smoothed max function. Note that unlike common adversarial example generation problems from the literature, our formulation is a minimization problem because of how we defined the objective. We solve (6) using projected gradient descent (Goldstein et al., 2014) in which each iteration updates the perturbation using Adam (Kingma & Ba, 2014), and then clips the perturbation to ensure that the $\ell_\infty$ constraint is satisfied.

---

[2]The first maxpool layer's output is always greater than or equal to the output of the second maxpool layer.

### 4.5 REMIX ADVERSARIAL EXAMPLES

The optimization problem defined in equation 6 tries to create an adversarial example with a fingerprint that does not look like the original signal's fingerprint. While this approach can trick the search algorithm used in copyright detection systems by lowering its confidence, it can result in unnatural sounding perturbations. Alternatively, we can try to enforce the perturbed signal's fingerprint to be similar to a different audio signal. Due to the approximate linear superposability characteristic of the spectrogram peaks, this will make the adversarial example sound more natural and like the target signal audio.

To achieve this goal, we will first introduce a loss function that tries to make two signals look similar rather than different. As described in equation 7, such a loss can be obtained by replacing the order of max over big and small neighborhoods in equation 4. Note that we will still use the smooth maximum from equation 5.

$$J_{remix}(x, y) = \sum_i \left( ReLU \left( c - \left( \max_{|j| \le w_2} \phi(i+j; x) - \max_{|j| \le w_1} \phi(i+j; x) \right) \right) \cdot \psi(i; y) \right) \quad (7)$$

Using this loss function, we define the optimization problem in equation 8, which not only tries to make the adversarial example different from the original signal $x$, but also forces similarity to another signal $y$.

$$\min_\delta J(x + \delta, x) + \lambda J_{remix}(x + \delta, y) \quad s.t. \quad \|\delta\|_p \le \epsilon. \quad (8)$$

Here $\lambda$ is a scale parameter that controls how much we enforce the similarity between the fingerprints of $x + \delta$ and $y$. We call adversarial examples generated using equation 8 "remix" adversarial examples as they sound more like a remix, and refer to examples generated using equation 6 as default adversarial examples. While a successful attack's adversarial perturbation may be larger in the case of remix adversarial examples (due to the additional term in the objective function), the perturbation sounds more natural.

## 5 EVALUATING TRANSFER ATTACKS ON INDUSTRIAL SYSTEMS

We test the effectiveness of our black-box attacks on two real-world audio search/copyright detection systems. The inner workings of both systems are proprietary, and therefore it is necessary to attack these systems with black-box transfer attacks. Both systems claim to be robust against noise and other input signal distortions.

We test our system on a dataset containing the top billboard songs from the past 10 years. We extract a 30-second fragment of these songs and craft both our default and remix adversarial examples for them. Although both types of adversarial examples can dodge detection, they have very different characteristics. The default adversarial examples (equation 6) work by removing identifiable frequencies from the original signal, while the remix adversarial examples (equation 8) work by introducing new frequencies to the signal that will confuse the real-world systems.

### 5.1 WHITE-BOX ATTACK RESULTS

Before evaluating black-box transfer attacks against real-world systems, we evaluate the effectiveness of a white-box attack against our own proposed model. Doing so will allow us to have a baseline of how effective an adversarial example can be if the details of a model are ever released or leaked.

To create white-box attacks against our model, we use the loss function defined in equation 3. By optimizing this function, we can remove almost all of the fingerprints identified by our model with perturbations that are unnoticeable by humans. Table 1 shows the norms of the perturbations required to remove 90%, 95%, and 99% of the fingerprint hashes.

### 5.2 TRANSFER ATTACKS ON AUDIOTAG

AudioTag[3] is a free music recognition service with millions of songs in its database. When a user uploads a short audio fragment on this website, AudioTag compares the audio fragment against a

---

[3]https://audiotag.info/

| Percentage of removed hashes | 90% | 95% | 99% |
|---|---|---|---|
| Perturbation norm ($\ell_\infty$) | 0.012 | 0.023 | 0.038 |
| Perturbation norm ($\ell_2$) | 0.004 | 0.005 | 0.006 |

Table 1: Norms of the perturbations for white-box attacks. Before computing the norms, we have normalized the signals to have samples that lie in $[0, 1]$.

database of songs and identifies what song this audio fragment belongs to. AudioTag claims to be "robust to sound distortions, noises and even speed variation, and will therefore recognize songs even in low quality audio recordings".[4] Therefore, one would expect that low-amplitude non-adversarial noise should not affect this system.

As shown in Figure 2, AudioTag can accurately detect the songs corresponding to the benign signal. However, the system fails to detect both the default and remix adversarial examples built for them. During our experiments with AudioTag, we realized that this system is relatively sensitive to our proposed attacks and it can be fooled with relatively small perturbation budgets. Qualitatively, the magnitude of the noise required to fool this system is small and it is not easily noticeable by humans. Based on this observation, we suspect that the architecture of the fingerprinting model used in AudioTag may have similarities to our surrogate model in section 4.2.

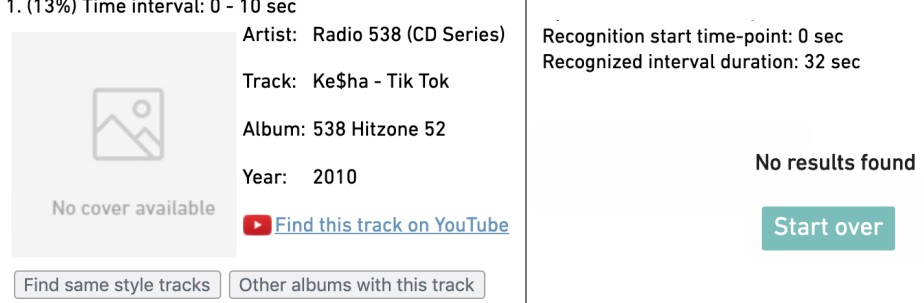

Figure 2: AudioTag can identify the benign audio signals, but fails to detect adversarial examples.

Table 2 shows the $\ell_\infty$ and $\ell_2$ norms of the perturbations required to fool AudioTag on 90% of the songs in our dataset. We also verified AudioTag's claim of being robust to input distortions by applying random perturbations to the audio recordings. To fool AudioTag with random noise, the magnitude ($\ell_\infty$) of the noise must be roughly 4 times larger than the noise we craft using equation 6.

| Target model | AudioTag | | | YouTube | | |
|---|---|---|---|---|---|---|
| Type of perturbation | default | remix | random noise | default | remix | random noise |
| Perturbation norm ($\ell_\infty$) | 0.03 | 0.03 | 0.12 | 0.10 | 0.10 | 0.32 |
| Perturbation norm ($\ell_2$) | 0.02 | 0.02 | 0.06 | 0.07 | 0.08 | 0.19 |

Table 2: Norms of the perturbations in adversarial examples that can evade each real-world system. Before computing the norms, we have normalized the signals to $[0, 1]$.

## 5.3 YOUTUBE

YouTube[5] is a video sharing website that allows users to upload their own video files. YouTube has developed a system called "Content ID[6]" to automatically tag user-uploaded content that contains copyrighted material. Using this system, copyright owners can submit their content and have YouTube scan uploaded videos against it.

As shown in the screenshot in Figure 4, YouTube Content ID can successfully identify the benign songs we use in our experiment. At the time of writing this paper both our default and remix attacks

---

[4]https://audiotag.info/faq

[5]https://www.youtube.com/

[6]https://support.google.com/youtube/answer/2797370?hl=en

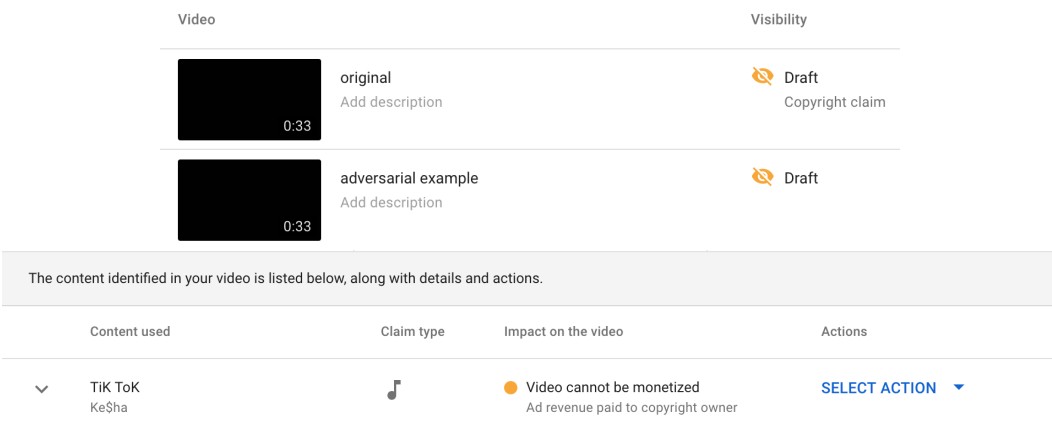

Figure 4: YouTube can successfully identify the benign/original audio signal while it fails to detect the adversarial examples.

successfully evade Content ID and go undetected. However, YouTube Content ID is significantly more robust to our attacks than AudioTag. To fool Content ID, we had to use a larger value for $\epsilon$. This makes perturbations quite noticeable, although songs are still immediately recognizable by a human. Furthermore, a perturbation with non-adversarial random noise must have an $\ell_\infty$ norm 3 times larger than our adversarial perturbations to successfully avoid being detected.

We repeated our experiments with identical hyper-parameters on the songs from our dataset. Table 2 shows the $\ell_\infty$ norms (i.e., the parameter $\epsilon$) and $\ell_2$ norms of the perturbations required to fool YouTube on 67% of the songs in our dataset. Furthermore, figure 3 shows the recall of YouTube's copyright detection tool on our dataset for different magnitudes of perturbations.

## 6 CONCLUSION

Copyright detection systems are an important category of machine learning methods, but the robustness of these systems to adversarial attacks has not been addressed yet by the machine learning community. We discussed the vulnerability of copyright detection systems, and explain how different kinds of systems may be vulnerable to attacks using known methods. As a proof of concept, we build a simple song identification method using neural network primitives and attack it using well-known gradient methods. Surprisingly, attacks on this model transfer well to online systems.

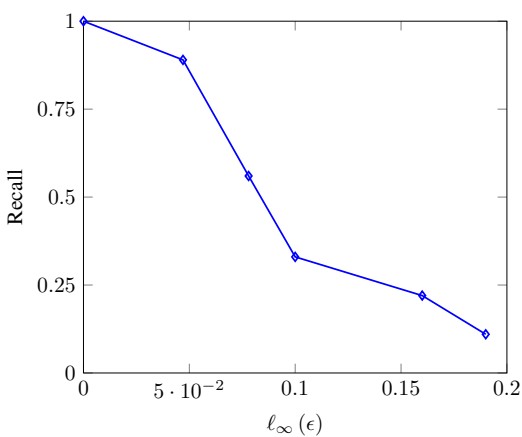

Figure 3: YouTube's copyright detection recall against the magnitude of noise on top Billboard songs dataset

Note that none of the authors of this paper are experts in audio processing or fingerprinting systems. The implementations used in this study are far from optimal, and we expect that attacks can be strengthened using sharper technical tools, including perturbation types that are less perceptible to the human ear. Furthermore, we are doing transfer attacks using fairly rudimentary surrogate models that rely on hand-crafted features, while commercial system likely rely on full trainable neural nets.

Our goal here is not to facilitate copyright evasion, but rather to raise awareness of the threats posed by adversarial examples in this space, and to highlight the importance of hardening copyright detection and content control systems to attack. A number of defenses already exist that can be utilized for this purpose, including adversarial training.

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
