# OpenReview forum: "Adversarial Attacks on Copyright Detection Systems"
_ICLR.cc/2020/Conference — Reject_

### Official Review · AnonReviewer1 · 2019-10-24
**Official Blind Review #1**

**Rating:** 3

**Review:**

This paper points out that the industrial copyright detection systems are susceptible to adversarial attacks and discuss the reason why these systems, especially the neural-network-based systems, are vulnerable to attacks. The authors describe a widely used music identification method and propose a simple gradient method to attack this system in this paper.  Specifically, the authors build a simple song identification method using neural network primitives and propose a differentiable surrogate loss to measure how close two fingerprints of audio are, which is used to generate the adversarial examples. And then they attack a song identification using the adversarial examples.
The authors conduct experiments to evaluate the proposed method. The white-box attack on their own model and the experiments on two real-world systems (AudioTag and YouTube) validate the effectiveness of the proposed method. The experimental results on AudioTag and YouTube seem impressive.

Weak points:
From my point of view, this paper is more like an application-oriented paper than a research paper. Because the main contribution of this paper is the results of experiments on a real-world system. Though the experimental results seem to support the justification of the model, this paper has limitations in terms of Model and Experiments.
1.	Model
a.	The description of the proposed model is not clear enough. The whole architecture of this model is not presented. The authors should describe how to generate the adversarial example more clearly.
b.	In section 4.1.2, “we build a shallow neural network that captures the key ideas of Wang et al. [2003] while adding extra layers that help produce transferable adversarial examples”. The Audio fingerprinting models proposed in the paper are quite incremental.
c.	The proposed differentiable surrogate loss, which measures how close two fingerprints of audio are, should have more theoretical analysis to illustrate its reasonability or provide some analysis of the difference between adversarial examples and real examples.
2.	Experiment
a.	The results of the paper are hard to reproduce.  The detail of the model used in the experiment is not presented, such as how to train the model.
b.	The authors should add more baseline or other intuitive methods to make the results of both the white-box attack and real-world more convincing.


**Experience Assessment:**

I do not know much about this area.

**Review Assessment: Checking Correctness Of Derivations And Theory:**

N/A

**Review Assessment: Checking Correctness Of Experiments:**

I assessed the sensibility of the experiments.

**Review Assessment: Thoroughness In Paper Reading:**

I read the paper at least twice and used my best judgement in assessing the paper.

---

> ### Author Response · Authors · 2019-11-14
> **Response to R1**
>
> Thank you for taking the time to review our work.
>
> First, we would like to point out that as mentioned in the general response, this is intended to be an applications paper. The main contribution of this paper is neither building a new fingerprinting model (we used the model proposed by Wang et al., with an extra normalization layer that promotes transferability of our adversarial examples to black-box models), nor proposing a novel method for creating adversarial examples. Instead, this work is mainly focused on studying why copyright systems are susceptible to adversarial attacks, and quantifying this susceptibility.
>
> There is a common belief in industry that adversarial attacks are not a real threat to real-world systems and there is no need for having robust machine learning models in production environments.  We believe this perception is largely the result of the extremely narrow focus of the ML security community on toy image classifiers (usually only MNIST and CIFAR) and stop sign detectors (which are often identified by autonomous vehicles using GPS rather than vision). While there is no doubt about the importance of such works in advancing our knowledge about adversarial attacks, they do not convince others with engineering and applied research backgrounds that adversarial attacks can happen in sophisticated industrial systems. Therefore, the main focus of this paper is to raise awareness among the broader machine learning community that adversarial attacks are real, they can happen in real-world scenarios, and as a society we are not prepared for them.
>
> That being said, we would like to point out the novelty of the loss function defined in equation (4). There are no other works that have been able to create a successful attack against audio copyright detection systems at this time, and numerous technical challenges had to be overcome to fool industrial systems with a perturbation 4X smaller than random noise using such a simple surrogate model.
>
>
> Regarding your comment on training of the model, we would like to point out that the model we used in this work does not have any unknown variables that need to be trained - it relies entirely on hand crafted features. For the hyper parameters of the model we used in this work, we tried to use values most commonly used by others. More specifically, we used a window size of 4096 with an overlap ratio of 0.5 for the short term Fourier transform layer, and a window size of 11 for the Hann function.
>
> Regarding adding more baseline methods, we would like to point out that there are no prior works that have tried to create adversarial examples for copyright detection models. While the random noise was the most intuitive baseline that we could come up with, we would happily include other baselines if you have any specific baselines in mind.
>
> In the end, we would like to thank you again for your time and your insightful comments. We hope that we had been able to address your concerns.

---

### Official Review · AnonReviewer2 · 2019-10-29
**Official Blind Review #2**

**Rating:** 3

**Review:**

Thank you to the authors for clarifying that they disclosed the vulnerability to YouTube. It would be good to surface this in the main text rather than a footnote, and to include some details as to what the response from YouTube was.

The paper demonstrates that adversarial examples are a practical scheme for evading copyright infringement detection tools. The study of this application is well motivated: adversarial examples need not be played in the physical domain (since the audio/video is uploaded directly in a digital format) and copyright detection is a bit more complicated than object classification (because false positives are expensive), small margin between classes.

Could you motivate the choice of the L_inf norm as an appropriate metric for the magnitude of the perturbation added to audio samples? Would the perturbation affect a listener’s ability to follow the audio that the attacker is trying to avoid detection of? In Section 4.5, why is inserting a perturbation that is close to a different audio sample a successful attack? How would the baseline of playing the target audio sample at a very low volume perform in comparison?

While the paper considers a system that was designed by the authors, because they demonstrate transferability to commercial products, this is not a limitation of their work and demonstrates a realistic threat model. The description of experimental results is however missing important details required to evaluate the results presented:

* What are the details of the dataset used to evaluate the approach? How many songs were included in the dataset?
* Did you conduct a study to verify that the perturbation added did not affect legitimate listeners?

The paper is well written and easy to follow.

A simple nitpick on page 2: do you have any references to point to for the following claim? “Most audio, im- age, and video fingerprinting algorithms either train a neural network to extract fingerprint features, or extract hand-crafted features.”

And on page 7, the following sentence might be a typo given that the perturbations introduced in this paper are adversarial: “Therefore, one would expect that low-amplitude non-adversarial noise should not affect this system.”

**Experience Assessment:**

I have published in this field for several years.

**Review Assessment: Checking Correctness Of Derivations And Theory:**

I assessed the sensibility of the derivations and theory.

**Review Assessment: Checking Correctness Of Experiments:**

I carefully checked the experiments.

**Review Assessment: Thoroughness In Paper Reading:**

I read the paper thoroughly.

---

> ### Author Response · Authors · 2019-11-14
> **Response to R2**
>
> Thank you for taking the time to review our work.
>
> We believe your comment on the use of L_inf norm as the measure of magnitude of perturbation is valid in the sense that this norm is probably not the best measure for what a human's ears can detect. The choice for L_inf norm was mostly motivated by 1) use of this norm as the measure for magnitude of perturbation in existing works in adversarial machine learning, and 2) the relatively standard optimization of the min-max loss with this norm.
>
> Regarding the success of the proposed attack in section 4.5, our hypothesis is that this attack creates noise artifacts in the song that are very different than what a random or environmental noise would sound like, and survive through any smoothing measure that a black-box model may have. However, this is only our guess and we can't confidently say how these examples are affecting the internal mechanisms of the black-box models.
>
> Regarding the baseline of creating low volume audio samples, our experience is that the model proposed by Wang et al. (which is one the most basic audio fingerprinting model) is resilient against such an attack.
>
> The dataset we used for this work contained 10 songs: the top billboard song from each of the last 10 years. Regarding the size of this dataset, we would like to point out the difficulty in obtaining such a dataset. For a song to be flagged by YouTube it must be copyrighted, which means it won't be part of any publicly available dataset.
>
> While we did not conduct an official user study regarding the perceived quality of the adversarial examples, we are confident of the outcome of such a study:  white box attacks are imperceptible (even the authors can't hear them under good listening conditions), and black-box perturbations (at least using our simple surrogate model) are clearly perceptible [sample of black-box attacks on non-copyrighted song: https://github.com/advattackcopyright/samples/tree/master/Examples_supp].
>
> The claim we made on page 2 regarding that “Most audio, image, and video fingerprinting algorithms either train a neural network to extract fingerprint features, or extract hand-crafted features”, is just the result of a literature review we did. We might have very well missed some prior works.
>
> The sentence “Therefore, one would expect that low-amplitude non-adversarial noise should not affect this system” on page 7, refers to the claim made by AudioTag that their system is “robust to sound distortions, noises and even speed variation, and will therefore recognize songs even in low quality audio recordings”.
>
> Please see our general response as well.
>
>
> In the end, we would like to thank you again for your time and your insightful comments. We hope that we had been able to address your concerns.

---

### Official Review · AnonReviewer4 · 2019-10-30
**Official Blind Review #4**

**Rating:** 3

**Review:**

[Summary] This paper proposed a black(white)-box attacking algorithms to attack industrial copyright detectors. Specifically, a simple un-targeted gradient-based method can successfully fool commercial system like YouTube and AudioTag. This paper has generalized the concept of “adversarial examples” to a new application area, which is indeed a very important one, to raise awareness for the potential threats.

There are a couple of reasons of why I like this paper: (1) copyright attack and especially defense (missing in this paper) is a more important application compared to others widely studied vision tasks like classification. To increase the copyright detectors' accuracy, companies must have invested a huge amount of money on deep learning based methods. As this paper suggests, these more accurate detectors also come with severe security problems, and such problems are rarely studied before. In addition, the experimental results in this paper show that two commercial systems can be easily compromised in a black-box manner. (2) This paper is clear in writing and easy to follow. I am not an expert in acoustic/audio recognition, but this paper shows that a not very complicated approach can work in real-world without too much domain/expert knowledge.

My rating for this paper is “3: weak reject”. I think it’s below the line for two reasons: technical novelty and experimental settings. I will elaborate these two points in the following sections.

As the author mentioned and generally believed, almost every machine learning model (black and white-box) can be compromised with adversarial examples, especially in un-targeted manner. “Copyright detector” is just another machine learning model regardless its important usage. The technical approach proposed in this paper is straightforward and limited. First, section 4.2 is very similar to Wang et al. (2003). What’s the biggest improvement or novelty here? All the filters are handcrafted and fixed as suggested by Eq.1 and Eq.2. Why don’t train a network or take a pre-trained DNNs to predict spectrogram?  The biggest advantage of deep learning (end2end trainable as feature extractor) is not utilized.  Second, the solution of Eq.5 is award. Usually we don’t use a softmax with temperature to estimate max, better ways should be tried like gumbel-softmax. Third, besides the optimization methods used for Eq.6, Carlini&Wagner is also widely used for optimizing this objective. What’s the reason that C&W is missed in this paper? Fourth, Eq.8 can be used for targeted attacks. However, this part is missing. Un-targeted attack is considered as easy especially in white-box cases. Targeted attack is a better and harder task for evaluation of algorithms. Fifth, the surrogate loss Eq.4 is basically a hinge loss between two quantized feature maps with mask \psi. It’s pretty standard to me and I am surprised this is the only loss proposed and studied.

The experiments in this paper is more like a “proof-of-concept” rather than “serious evaluation”. First problem is that the norm is used to evaluate the perturbation. Unlike the norm in image domains which can be visualized and easily understood, I don’t have a clear understanding of how “big” the perturbations are for audios. A cognitive study or something like a user study should be conducted. Another question related to this, the random noise is 3x bigger in terms of norm, does this make huge difference when listening to it? Are these two perturbations both very obvious or both unnoticeable? Second, it seems like a dataset is built but the stats are missing. How big is this dataset? How many songs are used to generate results in Tab.1, Tab.2, and Fig.3? What’s the attack accuracy on AudioTag and Youtube? Third, no baseline methods are compared to in this paper, not even an ablation study. The proposed two methods (default and remix) seem to perform similarly. I don’t think the best thing people can do previously to attack copyright system is just random noise. The authors could argue that there are no previous papers, but I think more realistic baselines should be studied.

One question, the inline equation at the bottom of Page.4  should be \phi(x) == maxpool(\phi(x)) right? And \psi(x) = ( \phi(x) == maxpool(\phi(x)) ) ?

Overall, I think this paper is not in good shape to be published even though the targeted problem has already become a big concern right now.


**Experience Assessment:**

I have published one or two papers in this area.

**Review Assessment: Checking Correctness Of Derivations And Theory:**

I assessed the sensibility of the derivations and theory.

**Review Assessment: Checking Correctness Of Experiments:**

I carefully checked the experiments.

**Review Assessment: Thoroughness In Paper Reading:**

I read the paper thoroughly.

---

> ### Author Response · Authors · 2019-11-14
> **Response to R4**
>
> Thank you for taking the time to review our work.
>
> We would like to point you to our general response. Here we will discuss the remaining concerns raised:
>
> Regarding your comment on why we did not train a network to predict spectrogram, we would like to point out that the scope of this paper was not to develop a novel copyright detection model - the issue of how to best construct neural models for this purpose is an entire research field in its own right, and is outside out scope.  We chose to attack the model of Wang et al. because it is the only currently available example of an industrial system that has been published.
>
> While we agree with your comment regarding the other optimization methods for equation 6, we would like to point out that the goal of this paper was to create a proof of concept for creating adversarial examples for copyright detection models. However, if you believe that using other optimization methods would help further improve our proof of concept, we will happily include them in the next version of our submission.
>
> As you correctly pointed out, equation 8 was originally designed for targeted attacks. While with this equation we were able to create un-targeted examples the had different characteristics than the ones created from equation 6, we did not manage to create any targeted attacks. We believe this happened due to the black-box nature of our attacks.  We are creating examples for black-box models about which we have absolutely no information. We don't even know if the black-box models are using DNNs, or some other approach for fingerprinting.  Without even knowing the classes/outputs of their system, or its inner workings, it is a very high bar to execute a targeted attack.
>
> While the proposed loss in equation 4 is a relatively simple loss, we believe it perfectly captures our goal: to push the local maximum out of a neighborhood around it's original location. Furthermore, there is no previous baseline in the literature that to compare to.
>
> While we did not conduct an official user study regarding the perceived quality of the adversarial examples, we are confident of the outcome of such a study:  white box attacks are imperceptible (even the authors can't hear them under good listening conditions), and black-box perturbations (at least using our simple surrogate model) are clearly perceptible [sample of black-box attacks on non-copyrighted song: https://github.com/advattackcopyright/samples/tree/master/Examples_supp].
>
> The dataset we used for this work contained 10 songs: the top billboard song from each of the last 10 years. Regarding the size of this dataset, we would like to point out the difficulty in obtaining such a dataset. For a song to be flagged by YouTube, it must be copyrighted which means it won't be a part of any publicly available datasets.
>
> Regarding adding more baseline methods, we would like to point out that there are no prior works that have tried to create adversarial examples for copy right detection models. While the random noise was the most intuitive baseline that we could come up with, we would happily include other baselines if you have any specific baselines in mind.
>
> Regarding the equation on the bottom of page 4, you are correct to assume that the equal sign is indeed a comparison operator.

---

### Official Review · AnonReviewer3 · 2019-10-30
**Official Blind Review #3**

**Rating:** 6

**Review:**

The authors of this work bring to light the security vulnerabilities of copyright detection systems to (DL style) adversarial attacks. The work summarizes the basics of copyright detection systems for audio, and notes that recent methodologies for feature extraction incorporate neural networks, as opposed to hand-crafted features. Following which the authors embark on designing a simple copyright system based on a neural network. With the newly proposed system, it is shown that one can construct adversarial examples to obstruct copyright detection systems using differentiable programming. The constructed adversarial examples are shown to be successful in evading popular copyright detection systems for audio (YouTube Content ID, AudioTag) as black box attacks.

Overall the paper brings about an interesting and pressing issue in a timely manner that seems to be of broad interest to the security and ML community. The paper is very well written and I enjoyed reading it. Further, the construction of the adversarial objective and attacks seems novel. However, I am not as familiar with the literature with adversarial attacks for audio. Overall I am giving this work a weak accept, which I am willing to change if the authors provide additional insight into their experiments.

In particular, it is difficult for me to assess the success of the imperceptibility of the perturbation in the audio domain from l2 and l_infinity norms, so it is hard for me to judge whether such adversarial examples are competitive and useful. For this reason I would like the authors share excerpts from their attack experiments for multiple examples for the different results presented.



**Experience Assessment:**

I do not know much about this area.

**Review Assessment: Checking Correctness Of Derivations And Theory:**

I carefully checked the derivations and theory.

**Review Assessment: Checking Correctness Of Experiments:**

I assessed the sensibility of the experiments.

**Review Assessment: Thoroughness In Paper Reading:**

I read the paper thoroughly.

---

> ### Author Response · Authors · 2019-11-14
> **Response to R3**
>
> Thank you for taking the time to review our work.
>
> We have uploaded a sample audio of a non-copyrighted song which to the anonymized link: https://github.com/advattackcopyright/samples/tree/master/Examples_supp
> Note that this song is not one of songs from our dataset (which are copyrighted songs) but it has perturbations which are as large as those used for the black-box attacks. The white-box attacks are identical to the original song.
>
> In the end, we would like to thank you again for your time and your insightful comments. We hope that we had been able to address your concerns.

---

### Author Response · Authors · 2019-11-14
**General response**

We thank all the reviewers for their time and their insightful comments.  Since many of the reviewers had similar concerns, to prevent redundancy, we are posting a general response and responding to the less general concerns individually:

* We would like to point out that this is intended to be an applications paper. The main contribution of this paper is neither building a new fingerprinting model, nor proposing a novel method for creating adversarial examples. Instead, this work is mainly focused on studying why copyright systems -- which are among the most influential systems that we use in our day to day lives -- are susceptible to adversarial attacks. Strangely, copyright detection is one of the most ubiquitous uses of ML in industry, and yet it is largely overlooked by the security community in favor of less realistic target problems.

* We chose to attack the fingerprinting model of Wang et al. because it is the only extant example of an industrial fingerprinting system that has appeared in the public domain.  We think this system is interesting for two reasons.  First, while it is known that neural nets are highly susceptible to a range of attack, our attack on this systems shows that even hand-crafted feature extractors can still be attacks.  Second, it is an example of a real commercial system, rather than a toy system we created in a lab.  This is in contrast to most work on adversarial attacks, which tends to focus on toy problems - e.g., MNIST, CIFAR, and ImageNet classifiers (as opposed to detectors).

* Several reviewers asked why we chose to attack the Wang et al model, rather than doing some reverse engineering to construct a better representation of an industrial system (like Content ID), which likely doesn't rely on hand-crafted features.  We were also asked whether this kind of research is ethical.  We think it is perfectly ethical to show that a category of vulnerabilities exists, and to provide information for practitioners who may be susceptible.  In the future, someone may do some reverse engineering to create an attack that is considerably more damaging than our (fairly benign) proof of concept.  This is not a direction we want to explore; our goal here was to write an interesting paper that explores an important issue, not to create security vulnerabilities that immediately impact companies and their employees.

---

### Decision · Program_Chairs · 2019-12-19

**Decision:**

Reject

**Comment:**

This paper shows a case study of an adversarial attack on a copyright detection system. The paper implements a music identification method with a simple convolutional neural network, and shows that it is possible to fool such CNN with an adversarial learning. After the discussion period, two among three reviewers incline to the rejection of the paper. Although the majority of the reviewers agree that this is an interesting problem with an important application, they also find many of their concerns remain unaddressed. These include the generality of the finding as the current paper is more like a proof-of-concept that black/white-box attack can work for copyright system. The reviewers are also concerned that the technique solution/finding is not novel as it is very similar to prior work in other domains (e.g., image classification). One reviewer was particularly concerned about that the user study is missing, making it difficult to judge whether the quality of the modified audio is reasonable or not.